# ctDNA-Based Liquid Biopsy of Cerebrospinal Fluid in Brain Cancer

**DOI:** 10.3390/cancers13091989

**Published:** 2021-04-21

**Authors:** Laura Escudero, Francisco Martínez-Ricarte, Joan Seoane

**Affiliations:** 1Vall d’Hebron Institute of Oncology (VHIO), Vall d’Hebron University Hospital, 08035 Barcelona, Spain; lescudero@vhio.net; 2Vall d’Hebron Institut de Recerca (VHIR), Vall d’Hebron University Hospital, 08035 Barcelona, Spain; frmartinez@vhebron.net; 3Universitat Autotònoma de Barcelona (UAB), 08193 Bellaterra, Spain; 4CIBERONC (Centro de Investigación Biomédica en Red de Cáncer), Instituto de Salud Carlos III, 28029 Madrid, Spain; 5ICREA (Institució Catalana de Recerca i Estudis Avançats), 08010 Barcelona, Spain

**Keywords:** central nervous system malignancies, brain cancer, circulating tumour DNA, cerebrospinal fluid, liquid biopsies

## Abstract

**Simple Summary:**

The optimal treatment and management of patients with brain cancer depend on the molecular characteristics of their tumour. Since the tumour changes with time, it is, therefore, essential to characterise the tumour of each patient at the exact time of selecting the most suitable therapeutic strategy. However, obtaining a tumour biopsy for its characterisation is a risky and invasive procedure and, sometimes, not even feasible, leading to a lack of information about the tumour. These challenges can be overcome by using a liquid biopsy of cerebrospinal fluid. Brain cancer cells release DNA into the cerebrospinal fluid, and the analysis of the cell-free circulating tumour DNA can reveal the genetic profile of brain cancer in a relatively noninvasive manner. In this review, we revise the recent results in this field that show how circulating tumour DNA in cerebrospinal fluid can provide diagnostic and prognostic information, identify potential therapeutic targets, monitor the tumour response or resistance to treatment, and help to identify tumour relapse.

**Abstract:**

The correct characterisation of central nervous system (CNS) malignancies is crucial for accurate diagnosis and prognosis and also the identification of actionable genomic alterations that can guide the therapeutic strategy. Surgical biopsies are performed to characterise the tumour; however, these procedures are invasive and are not always feasible for all patients. Moreover, they only provide a static snapshot and can miss tumour heterogeneity. Currently, monitoring of CNS cancer is performed by conventional imaging techniques and, in some cases, cytology analysis of the cerebrospinal fluid (CSF); however, these techniques have limited sensitivity. To overcome these limitations, a liquid biopsy of the CSF can be used to obtain information about the tumour in a less invasive manner. The CSF is a source of cell-free circulating tumour DNA (ctDNA), and the analysis of this biomarker can characterise and monitor brain cancer. Recent studies have shown that ctDNA is more abundant in the CSF than plasma for CNS malignancies and that it can be sequenced to reveal tumour heterogeneity and provide diagnostic and prognostic information. Furthermore, analysis of longitudinal samples can aid patient monitoring by detecting residual disease or even tracking tumour evolution at relapse and, therefore, tailoring the therapeutic strategy. In this review, we provide an overview of the potential clinical applications of the analysis of CSF ctDNA and the challenges that need to be overcome in order to translate research findings into a tool for clinical practice.

## 1. Introduction

Central nervous system (CNS) malignancies affect both children and adults worldwide and are responsible for substantial morbidity and mortality. An epidemiological study of CNS cancer between 1990 and 2016 revealed that the age-standardised incidence rate has increased by 17.3% globally, with 330,000 incident cases and 227,000 deaths globally in 2016 [1].

CNS cancer consists of primary tumours and intracranial metastases. Most CNS tumours (>90%) occur in the brain, with the remaining located in the meninges, spinal cord and nerves. Depending on the anatomical region and the tumour type, the neurological signs and symptoms will vary and may include headaches, seizures, loss of vision, paralysis, speech disturbance, and motor deficits [2].

CNS tumours are diagnosed using neuroimaging techniques such as magnetic resonance imaging (MRI) or computed tomography; however, to obtain pathological information and molecular diagnosis, tumour biopsies are required. The treatment strategy for primary CNS tumours consists of either obtaining a biopsy or performing a surgical resection, combined, when appropriate, with postoperative radiotherapy and chemotherapy [3].

CNS tumour prognosis is diverse since there are distinct entities with different histopathological characteristics and molecular profiles. Therefore, characterising the tumour specimen is essential for accurate diagnosis and prognosis, as well as to identify potential therapeutic targets. To improve disease control, primary brain tumours or single nodule brain metastases are resected. However, obtaining tumour biopsies is not always possible due to their location, particularly when CNS tumours occur in vital regions such as the basal ganglia or the brain stem. In addition, patients with disseminated disease may not be eligible for such procedures [4,5,6].

In cases where tumour resection or obtaining a biopsy is possible, the sample obtained may not be representative of tumour heterogeneity [7,8,9], and, therefore, multiple sampling may be required to confirm the pathological diagnosis in some cases. Moreover, the analysis of the sample obtained only provides a static snapshot from the time of resection. It is important to monitor the patient’s response to treatment during the course of the disease, particularly to distinguish true disease progression from a pseudoprogression induced by treatment. Sometimes a new or enlarging area of contrast enhancement is observed, but it is not easy to assess whether it is the result of tumour growth or an inflammatory response [10]. This can be challenging when using conventional imaging techniques.

Tumours evolve over time, particularly under the selective pressure of therapy, which can result in the expansion of pre-existing resistant clones or the acquisition of *de novo* resistant alterations [7]. Thus, the genomic characteristics at relapse may differ from the genomic landscape at first occurrence. In some cases, treatment decisions at relapse are just based on the characteristics of the primary tissue obtained [11,12]. The known evolution of tumours and the absence of longitudinal tumour sampling may, therefore, lead to imprecise diagnosis and clinical management.

For these reasons, there is an urgent need to develop less invasive methods to identify and validate tumour biomarkers that provide real-time information to aid in diagnosing and monitoring CNS malignancies. Overall, this will help to adjust the therapeutic strategy and guide treatment decisions based on the current tumour profile and its burden.

An alternative to a tumour biopsy is a liquid biopsy (Figure 1). Liquid biopsies are emerging as noninvasive tools that can provide longitudinal information about the tumour genomic landscape and facilitate patient monitoring. It consists of the analysis of biomarkers, including circulating tumour cells, exosomes and circulating tumour nucleic acids that are present in bodily fluids such as blood, cerebrospinal fluid (CSF), urine and saliva [13,14].

In this review, we will discuss the potential applications of circulating tumour DNA (ctDNA) in CSF as a biomarker for CNS malignancies, the challenges that we need to overcome, and future perspectives for its implementation in the clinical setting.

## 2. Circulating Cell-Free DNA and Circulating Tumour DNA

Cells release DNA that then circulate in bodily fluids. The fraction of cell-free DNA (cfDNA) that is shed by cancer cells, presumably undergoing apoptosis or necrosis, is known as ctDNA and carries genomic alterations that can be detected using PCR-based or next-generation sequencing (NGS)-based methods [14,15,16,17].

Increased concentrations of cfDNA have been detected in pathological conditions like trauma, infection and cancer, or even other physiological conditions like exercise [18].

For patients with an extracranial disease, plasma ctDNA has been detected across different cancer types. However, blood may not be a suitable source from patients with CNS malignancies since ctDNA levels were infrequently detected in plasma [19,20]. ctDNA was detectable in the plasma of >75% of patients with advanced cancers, such as bladder, colorectal, gastroesophageal, ovarian, pancreatic, breast, melanoma, hepatocellular, and head and neck cancers, in contrast with <10% of glioma patients (2/27) [19].

The proportion of ctDNA in the blood is small and varies depending on tumour characteristics, including type, grade and burden [16,19]. In contrast, the total amount of ctDNA in CSF is increased, making it an ideal biofluid to characterise and monitor CNS cancer [20,21]. Interestingly, the levels of ctDNA in CSF may be influenced by tumour burden, tumour progression and anatomical location of the tumour, with regard to the proximity to CSF reservoirs [21,22,23].

## 3. Cerebrospinal Fluid as a Source of ctDNA

CSF is a clear bodily fluid secreted by the choroid plexus that is present in the subarachnoid space of the brain, the spinal cord and the central canal [24].

CSF is in direct contact with the brain parenchyma, and several studies have shown that CSF is a reliable source of cell-free ctDNA, providing advantages over plasma or serum for the analysis of CNS tumours [20,21].

Several studies have reported the ability to detect ctDNA in CSF of patients with CNS malignancies. Gene mutations and molecular alterations have been detected in the CSF DNA of CNS cancer patients [25,26,27,28,29,30], followed by genomic landscape characterisation of CSF ctDNA with the development of high throughput sequencing technologies [20,21,22,23,31,32,33,34,35,36].

CSF samples can be accessed through a lumbar puncture or obtained from the ventricles under certain circumstances. Patients with posterior fossa tumours tend to present with hydrocephalus, a condition in which the CSF accumulates within the cerebral ventricles and/or subarachnoid spaces [37,38,39]. A lumbar puncture is contraindicated in these patients, given the risk of brain herniation; therefore, CSF is obtained from the ventricles during procedures that are performed to alleviate intracranial pressure and drain the excess of CSF [38,40,41,42].

As part of the CNS staging criteria for certain brain tumours, a diagnostic lumbar puncture is routinely performed as standard of care for CSF cytology evaluation of CNS lymphoma, CNS metastasis, and medulloblastoma [43,44,45,46]. In these cases, the CSF samples collected as standard of care can be further used to characterise the ctDNA and provide information about the tumour [20,34,47].

## 4. Clinical Applications of the CSF ctDNA for CNS Malignancies

CNS tumours are a heterogeneous group of malignancies [48]. In most cases, tumour resection is required to reduce tumour burden and mass effect [49]. However, liquid biopsies can be used to complement histopathological diagnosis and are essential for those patients with inoperable tumours.

The monitoring of CNS malignancies is currently performed by imaging techniques; however, these are not sensitive for microscopic disease [50]. A complementary liquid biopsy of CSF could, therefore, be performed to aid clinical assessment by determining the response to treatment, differentiating pseudoprogression from true progression and tracking levels of residual disease. In addition, the genomic characterisation of CSF ctDNA can facilitate the identification of actionable genomic alterations that confer sensitivity or resistance to clinically available drugs and the detection of mechanisms of resistance at relapse.

The distinct clinical applications of the analysis of CSF ctDNA are discussed below for patients with distinct types of CNS cancer.

### 4.1. Diffuse Gliomas

#### 4.1.1. A Diagnostic and Prognostic Tool

Among diffuse gliomas, glioblastoma (GBM) is the most common malignant brain tumour in adults, with a 2-year survival of 18% and 5-year overall survival (OS) of 4% [51]. Providing an accurate molecular profile for diagnosis and prognosis is essential and can be achieved with a CSF liquid biopsy. The analysis of the mutational status of *IDH1*, *IDH2*, *ATRX*, *TP53*, *TERT*, *H3F3A* and *HIST1H3B* in CSF ctDNA facilitates the molecular diagnosis of diffuse gliomas and provides prognostic information in a relatively noninvasive manner [22]. In addition, *TERT* promoter mutations have been detected in the CSF ctDNA of GBM patients, and shorter OS of patients with high variant allele frequency (VAF) has been observed. The results from this pilot study suggested that VAF levels of the *TERT* promoter mutation could be a predictor of poor survival [52]. In more recent studies, CSF was obtained from lumbar punctures in glioma patients, and ctDNA was detected and was associated with disease burden, tumour progression, and adverse outcomes [20,23]. Moreover, most patients with detectable ctDNA had a negative cytopathologic analysis, and ctDNA was not detected in plasma [20,23].

Diffuse midline glioma (DMG) is a tumour entity characterised by a K27M mutation in either *H3F3A* or *HIST1H3B/C*; it is usually located in the brain stem, thalamus and spinal cord [48]. Within H3 K27M-mutant DMG, diffuse intrinsic pontine glioma (DIPG) is a rapidly growing tumour in the brain stem that typically arises in young children and is associated with poor survival [53]. The anatomical location of these tumours, the brainstem, makes them difficult and dangerous to biopsy. Importantly, H3 K27M mutations can be detected in the CSF ctDNA, facilitating diagnosis and opening the possibility of avoiding diagnostic surgical biopsies [22,54,55].

The molecular characterisation and understanding of DIPG biology have been improved from specimens obtained from rare diagnostic biopsies and postmortem tissue donations [56,57,58,59]. Indeed, the lack of surgical specimens can be overcome by the analysis of CSF ctDNA to aid in the management of patients with DIPG and contribute to the molecular study of this disease to accelerate research. An NGS panel of 68 genes commonly mutated in brainstem tumours was used to study a cohort of 57 patients with brainstem tumours, including 23 patients with DIPG. Mutations were detected in the CSF ctDNA of 82.5% of patients, and the presence of *H3F3A/HIST1H3B* mutations was correlated with poor OS while the *IDH1* mutation predicted better OS [54]. Moreover, longitudinal analysis of CSF samples offers the possibility of monitoring and allows the tumour evolution of this dismal disease to be studied.

#### 4.1.2. Monitoring and Therapeutic Strategies

The number of actionable genomic alterations for patients with primary brain tumours is limited. The most relevant biomarker for glioma is *MGMT* promoter methylation status. *MGMT* promoter methylation causes the loss of *MGMT* expression, and since it is involved in DNA repair by reversing DNA alkylation, *MGMT* promoter methylation renders cells more susceptible to temozolomide and is associated with longer survival [60,61,62,63]. *MGMT* promoter methylation was detected using methylation-specific PCR from genomic DNA extracted from the CSF of glioma patients, with higher sensitivity than from serum [64].

A potential biomarker for GBM is epidermal growth factor receptor (*EGFR*). *EGFR* amplification and EGFRvIII mutation have been detected in RNA within extracellular vesicles circulating in CSF [65]. This could be of high interest as a biomarker to predict response to future EGFRvIII-targeted therapies in GBMs.

Longitudinal analysis of CSF ctDNA from glioma patients showed the evolution of the cancer genome through the mutational changes detected [23].

### 4.2. Brain and Leptomeningeal Metastases

#### 4.2.1. CSF ctDNA Facilitates Diagnosis and Allows Tumour Genomic Characterisation

About 20–40% of patients with advanced-stage cancers of the lung, breast and melanoma develop brain metastasis, and approximately 5–8% of these patients are diagnosed with leptomeningeal metastasis. These are devastating diseases that carry a poor prognosis and are often resistant to treatment [66,67,68].

In addition to brain metastasis present in the brain parenchyma, malignant cells can seed the leptomeninges, causing leptomeningeal metastasis [67,69,70]. The diagnosis of leptomeningeal metastasis is based on clinical symptoms, MRI scans, and cytology analysis of CSF [71]. However, up to 20% of patients with positive clinical and radiographic signs presented false-negative CSF cytology [72,73]. Several studies have shown that ctDNA can be detected in the CSF of patients with negative cytology analysis [20,27,32,74]. Cytology has limited sensitivity, and the analysis of ctDNA can complement the diagnosis of leptomeningeal metastases.

Brain metastasis can present different genomic alterations compared to their primary extracranial tumour [12]. Several studies have shown that the analysis of CSF ctDNA enables the characterisation of the genomic complexity of CNS metastases, including intratumour heterogeneity, revealed with the identification of trunk and private genomic alterations. Moreover, the genomic landscape of CNS metastasis, including the brain lesion’s private alterations, was better represented from the ctDNA in the CSF than plasma [20,32,75]. Analysis of CSF ctDNA from a cohort of 26 patients with leptomeningeal metastases from non-small cell lung cancer (NSCLC) revealed their unique genetic profiles, including mutations in several driver genes, copy number variations (CNVs) in *MET*, *ERBB2*, *KRAS*, *ALK*, and *MYC*, and loss of heterozygosity in *TP53* [76].

#### 4.2.2. Patient Monitoring and Identification of Therapeutic Targets

There are several targeted therapies for brain metastases [77,78,79,80,81,82,83,84,85,86]. For EGFR-mutated brain metastases from NSCLC, first-, second- and third- generation EGFR tyrosine kinase inhibitors (TKIs) are available [87,88,89,90]. In addition, for NSCLC with ALK gene rearrangement, CNS penetration and therapeutic potential were exhibited by second-generation ALK inhibitors [84,85,86]. There are also targeted therapies for patients diagnosed with HER2+ breast cancer and melanoma patients, including BRAF and MEK inhibitors [82,83,91,92]. For the treatment of leptomeningeal metastases, targeted therapies for the aforementioned actionable genomic alterations may also be effective [93].

The availability of targeted therapies highlights the importance of the identification of actionable genomic alterations or resistance mutations in genes, including *EGFR*, *ALK*, *BRAF* and *HER2*, which have been detected from CSF ctDNA in several studies [20,21,31,32,94,95,96]. For example, EGFR–TKI resistance mutation *EGFR* T790M has been detected in the CSF ctDNA of lung cancer patients [94,97].

A study of 21 patients with brain metastasis from NSCLC compared the NGS results obtained from different samples to reveal the mutation pattern of driver genes for each patient. Mutations were detected in 95.2%, 66.7% and 39% of patients from CSF ctDNA, plasma ctDNA, and plasma circulating tumour cells, respectively. The most mutated gene was *EGFR,* followed by *KIT*, *PIK3CA*, *TP53*, *SMAD4*, *ATM*, *SMARCB1*, *PTEN* and *FLT3* (all >15%). For *EGFR* mutations, the detection rate was 57.1% (12/21) from CSF ctDNA, which, interestingly, was higher for patients with leptomeningeal (81.8%; 9/11) compared with brain parenchymal (30%; 3/10) metastases [98].

The analysis of CSF ctDNA can also contribute to monitoring response to treatment. Metastasis in the CNS developed in a patient with HER2+ breast cancer. Analysis of baseline CSF ctDNA revealed mutations in *TP53* and *PIK3CA* and amplification in *ERBB2* and c*MYC*. Following treatment with T-DM1, extracranial disease control was achieved, and marker levels in plasma decreased. However, the levels increased in CSF ctDNA, consistent with poor treatment benefit to the CNS [99].

Altogether, these results indicate that CSF is a more suitable fluid than plasma to reveal the mutational profile of CNS metastases and can aid in diagnosis, tailored treatment selection and monitoring.

### 4.3. CNS Lymphoma

Malignant B-cells can infiltrate the CNS and are associated with poor prognosis, particularly at relapse [100]. Primary CNS lymphoma is defined by the absence of systemic disease in contrast to secondary CNS lymphoma that presents infiltration into the CNS with previous or concomitant systemic lymphoma [101,102].

ctDNA has only been detected in the plasma of a minority of patients with restricted CNS lymphoma [103,104]. In contrast, several studies of patients with CNS lymphomas detected ctDNA in CSF [47,105,106,107,108,109].

Diagnosis and monitoring of CNS lymphoma are challenging, given the difficulties of tumour biopsies and the lack of sensitivity of CSF standard analysis (cytology and flow cytometry) and neuroimaging. The detection of the *MYD88* L265P mutation strongly suggests the diagnosis of primary CNS lymphoma, and this mutation has been detected in the CSF ctDNA of patients with CNS lymphoma [47,106,108,109], showing that the analysis of CSF ctDNA could complement the diagnosis.

An NGS-based analysis of the CSF cfDNA of 8 patients with CNS lymphoma, at recurrence, detected tumour-derived genetic alterations and showed that the clearance of ctDNA from CSF was associated with sustained tumour responses [105].

The comparison of CSF ctDNA with plasma ctDNA and CSF standard analysis (cytology and flow cytometry) revealed that the analysis of CSF ctDNA better detected CNS disease in patients with B-cell lymphoma [47]. Moreover, longitudinal analysis of CSF ctDNA levels allowed the monitoring of response to treatment, the detection of residual disease and predicted relapse. The dynamic changes observed in CSF ctDNA recapitulated the evolution of the disease for patients with CNS lymphoma [47].

### 4.4. Medulloblastoma

CNS tumours are the leading cause of cancer-related mortality in children and adolescents due to the aggressiveness of certain subtypes, including medulloblastoma and high-grade gliomas such as DMG [110,111].

Medulloblastoma (MB), an embryonal tumour of the CNS, is the most aggressive brain tumour in childhood that can also occur in adults, although this is less common [112]. MB is a complex and evolving heterogeneous disease that can be divided into four molecular consensus subgroups (WNT, SHH, Group 3 and Group 4), with further subtypes identified [113,114,115,116,117]. The lack of sufficient sample, intratumour heterogeneity or the presence of disseminated disease make diagnosis and monitoring difficult [8,118,119,120,121]. However, hydrocephalus is common amongst these patients, and CSF samples can be obtained prior to tumour surgical resection or biopsy [40,42]. In addition, CSF samples are routinely collected through a lumbar puncture for cytology analysis to assess metastatic dissemination according to Chang’s M-staging system, in combination with brain and spinal MRIs [45].

The study of paediatric patients with MB showed that ctDNA was more abundant in CSF (76.9%, 10/13 patients) than plasma (1/13 patients) for patients with negative CSF cytology results. Moreover, exome sequencing of CSF ctDNA recapitulated the tumour mutational burden and the genomic alterations, including MB common mutations (*PTCH1*, *TP53*), CNVs (*MYCN* and *GLI2* amplification) and arm-level chromosomal aberrations (chromosome 17p loss), providing diagnostic and prognostic information [34]. Longitudinal CSF samples were also collected, and ctDNA analysis detected residual disease, identified intratumour and interlesion heterogeneity, and revealed a genomic transformation of the tumour at relapse [34]. More recently, another study reported the detection of ctDNA in CSF; however, shared genetic mutations between CSF and the tumour specimen were only identified in 22% (2/9) of patients. The authors suggested that this could be explained by the time-interval differences between tumour and CSF collection [122].

MB also presents abnormal DNA methylation changes, with distinct epigenetic signatures identified across MB subtypes that can be altered during tumour progression and treatment [114,123]. The epigenetic analysis of CSF ctDNA from 4 MB patients (3 with matching tumour samples) was attained. A positive correlation of tumour and CSF samples was identified, suggesting that CSF ctDNA could be used to monitor changes in MB tumour DNA methylomes and hydroxymethylomes. In addition, DNA methylation markers of diagnostic and prognostic value could be detected in the CSF ctDNA [124]. In summary, CSF ctDNA analysis could facilitate the clinical management of paediatric patients with MB.

## 5. Challenges and Limitations

Obtaining CSF samples is less invasive than surgery; however, in some cases, a lumbar puncture is not feasible. Contraindications to performing a lumbar puncture include risk for cerebral herniation, abnormal intracranial pressure and coagulation abnormalities [41].

The analysis of ctDNA could be used as a biomarker of residual disease. However, it will be important to establish the dates of sample collection, particularly postsurgery, since the abundance of trauma-induced cfDNA, up to 4 weeks from the surgical procedure, could dilute the fraction of ctDNA and influence the results [125].

Another limitation is that ctDNA is not detected in all patients with CNS malignancies. Detection of ctDNA may be influenced by tumour burden, tumour progression and anatomical location [21,22,23]. Therefore, further research to determine the biological factors involved and improve technological sensitivity will be required.

For patient monitoring through the analysis of specific mutations by sensitive techniques such as droplet digital PCR (ddPCR), prior knowledge of the tumour genetic profile is required, and new mutations (not present in the primary sample) can appear in the relapse setting. In contrast, whole-exome sequencing (WES), shallow whole-genome sequencing (WGS) or specific gene panels might provide more information to aid with tumour characterisation and monitor residual disease. However, it is important to consider sensitivity, turnaround time and cost-effectiveness.

In addition, imaging techniques may sometimes reflect either inflammatory processes from treatment or neoplastic progression [10]. Further research is needed to investigate whether the analysis of CSF ctDNA can help distinguish between true progression from pseudoprogression.

To determine the impact of the results and translate them into a tool for clinical practise, standardisation of protocols and larger studies with more patients will be required. The implementation of well-designed and controlled clinical trials will be essential to validate the use of CSF ctDNA as a liquid biopsy for the clinical management of patients.

## 6. Conclusions and Future Insights

CNS cancer is a dismal disease. It has elevated mortality and disabling effects on patients and is a massive burden on global health care systems. However, early detection and treatment can result in improved outcomes [1].

Several studies of the CSF ctDNA of patients with different types of primary and metastatic CNS tumours have been performed and show promising results, highlighting the potential of CSF ctDNA as a biomarker.

The challenges will be to translate these findings into clinically validated assays to improve patient healthcare. Standardisation of protocols and further studies and clinical trials will be necessary to translate the current results into a feasible tool for its implementation in the clinical setting.

A liquid biopsy of CSF can characterise the tumour for diagnosis and provide prognostic information, also complementing the information obtained from the tumour sample if a biopsy or resection is feasible (Figure 1). During patient follow-up, it could be used to monitor the response to treatment through the levels of ctDNA and the detection of minimal residual disease. Importantly, it can facilitate early detection of relapse and identify therapeutic targets or mechanisms of resistance in order to adjust the therapeutic strategy at relapse (Figure 2).

Altogether, the analysis of CSF ctDNA remains a promising strategy to improve the clinical management of patients with CNS malignancies, and further studies are required to make liquid biopsies a standard clinical tool.

## Figures and Tables

**Figure 1 cancers-13-01989-f001:**
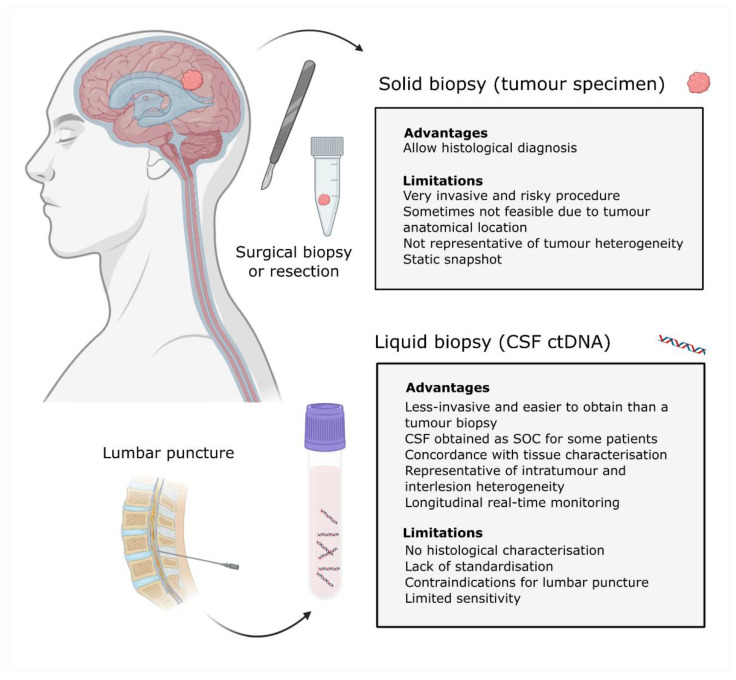
Solid vs. liquid biopsies. Schematic representation of the tumour biopsy and CSF samples obtained from a patient with a CNS malignancy. Advantages and limitations for each methodology are indicated. Definition of acronyms: standard of care (SOC), cerebrospinal fluid (CSF), circulating tumour DNA (cfDNA) and central nervous system (CNS).

**Figure 2 cancers-13-01989-f002:**
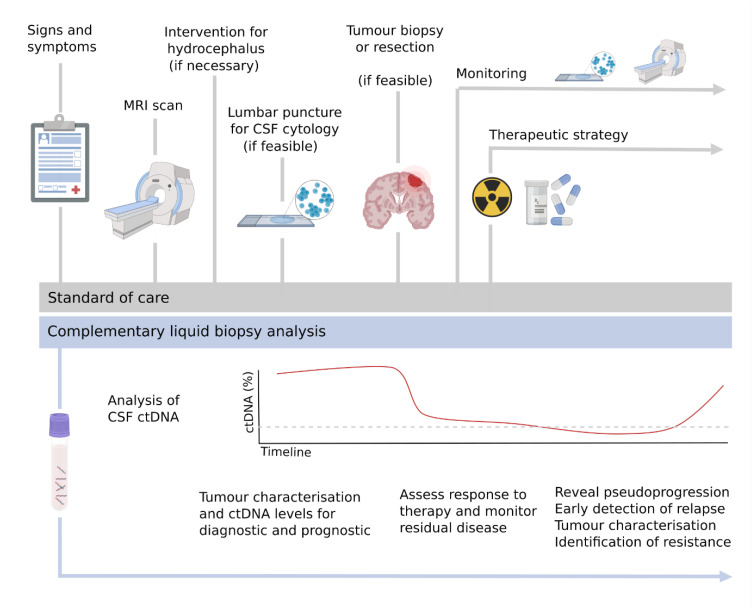
Liquid biopsies in the clinical setting. Schematic representation of the standard of care for patients with CNS malignancies and the complementation with a longitudinal analysis of CSF ctDNA.

## Data Availability

No new data were created or analyzed in this study. Data sharing is not applicable to this article.

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
