# Peer review of "ctDNA-Based Liquid Biopsy of Cerebrospinal Fluid in Brain Cancer"

_cancers, 2021, doi:10.3390/cancers13091989_

Round 1
Reviewer 1 Report
This is a well-written overview regarding CSR as source of information with regard to intracerebral tumours/metastases.
Data are presented in a balanced way.
Author Response
We thank the reviewer for his/her comments and for revising the manuscript.
Reviewer 2 Report
The authors have described the use of liquid biopsy as a non-invasive complementary analysis in tumor cancer and have nicely summarized the current studies of cerebrospinal fluid ctDNA in diagnosis and prognosis of different central nervous system tumor types. It also provides an overview of achievement as well as challenges that might allow the application of CSF ctDNA in clinics, which facilities the following studies in this area.
Minor point:
1. Reference on line 269 is not in the same format as other references.
Author Response
We thank the reviewer for revising the manuscript and identifying the missing reference. The reference has been introduced in the appropriate format.
Reviewer 3 Report
The presented review by Escudero et al. stresses that extracting cell free tumour DNA from the cerebrospinal fluid of patients with CNS tumours, can serve as a tool to characterize tumour heterogeneity and eventually monitor tumour progression as well as susceptibility to treatment.
I found the review to be highly interesting. However, I have some comments regarding the structure and content of the review:
- The review is not well structured in my opinion. Section 2 should be a part of the Introduction and not presented as a section on its own. Sections 6-9 should be sub-sections of section 5 which deals with clinical applications of the technique. Additionally, the text should be categorized according to application (diagnosis, biomarker for susceptibility to treatment, etc.) and not type of malignancy. Several types of malignancies can be presented under one application.
- Some technical data is missing. For example, how is tumour DNA differentiated from total cell-free DNA? What are the sequencing techniques that are usually applied? How is methylation of guanine detected?
- The use of abbreviations for technical terms in the text is too frequent and generates a feeling as though the text is written in code. Abbreviations should be used for terms that are repeated throughout the text, and not for ones that are only briefly mentioned and can therefore be written in full.
- I feel that adding a short overview of how the technique was demonstrated in pre-clinical models of CNS tumours would strengthen the review.
- Line 51: "For intracranial tumours, obtaining a tumour specimen is challenging since surgical resections are invasive and risky procedures." This sentence is redundant since its concept is repeated in the following lines.
- Line 63: "This can be challenging with conventional imaging techniques." Please briefly explain the shortcomings of medical imaging in distinguishing true from pseudo tumour progression.
- Please explain the term "the selective pressure of therapy" that appears in line 65.
- Line 99: " The detectable levels of ctDNA in plasma for patients with primary brain tumours may be influenced by the blood-brain-barrier (BBB) integrity". Please briefly elaborate on the selective permeability induced by the BBB and why is tumour DNA impermeable. Importantly, why doesn’t the presence of tumour vasculature, which is much less selective, result in high plasma levels of cell-free, tumour DNA?
- In line 121, please explain the circumstances in which CSF is extracted directly from the ventricles rather than by lumbar puncture? Please add references if possible.
- Lines 124-126: " To alleviate the intracranial pressure and to drain the excess CSF, procedures including ventriculoperitoneal shunt (VPS), ventricular drain (EVD) and endoscopic third ventriculostomy (ETV) are performed." This sentence seems irrelevant to the concept of the review.
- Is there a benefit to a liquid biopsy in cases where the tumour can be surgically resected? Or will the solid biopsy provide all the required information? Please elaborate.
- Please explain the term "longitudinal analysis".
- Please explain the term "actionable genomic alterations".
- In lines 184-186, please clarify that MGMT promoter methylation increases susceptibility to temozolomide. The reader may mistakenly think that MGMT expression does that.
- Section 4 deals more with ctDNA in the CSF that with the concept of the CSF in general. Therefore, it should be titled otherwise.
- In lines 255-257, in how many patients was ctDNA found?
- In line 269, the cited reference is not in the appropriate format.
Author Response
1- The review is not well structured in my opinion. Section 2 should be a part of the Introduction and not presented as a section on its own. Sections 6-9 should be sub-sections of section 5 which deals with clinical applications of the technique. Additionally, the text should be categorized according to application (diagnosis, biomarker for susceptibility to treatment, etc.) and not type of malignancy. Several types of malignancies can be presented under one application.
We thank the reviewer for his/her point; we have now included section 2 in the introduction. However, we would like to ask to maintain the original structure regarding sections 6-9. We prefer to highlight the distinct applications of the analysis of CSF ctDNA focusing on distinct central nervous system tumours for which information has been reported.
2- Some technical data is missing. For example, how is tumour DNA differentiated from total cell-free DNA? What are the sequencing techniques that are usually applied? How is methylation of guanine detected?
We have incorporated the following sentence (underlined) and a reference that provides more information on methods to detect ctDNA (Siravegna et al., 2019)
“Cells release DNA that then circulate in bodily fluids. The fraction of cell-free DNA (cfDNA) that is shed by cancer cells, presumably undergoing apoptosis or necrosis, is known as ctDNA and carries genomic alterations that can be detected using PCR-based or next generation sequencing (NGS)-based methods [14-17]” (REFS +Siravegna et al.,2019)
Moreover, as indicated within the text (322-328), techniques used for the analysis of ctDNA include droplet digital PCR (ddPCR), whole exome sequencing (WES), shallow whole genome sequencing (WGS) or specific gene panels.
ctDNA is differentiated from cfDNA by the detection of genomic alterations (mutations, copy number alterations and chromosomal rearrangements). This is explained in the text as “The fraction of cell-free DNA (cfDNA) that is shed by cancer cells, presumably undergoing apoptosis or necrosis, is known as ctDNA and carries genomic alterations”
In Wang et al., MGMT promoter methylation was detected using methylation-specific PCR from genomic DNA extracted from glioma tumor tissue, corresponding serum and CSF samples. This is further clarified in the text (underlined):
“MGMT promoter methylation was detected using methylation-specific PCR from genomic DNA extracted from the CSF of glioma patients with higher sensitivity than from serum [61].”
3- The use of abbreviations for technical terms in the text is too frequent and generates a feeling as though the text is written in code. Abbreviations should be used for terms that are repeated throughout the text, and not for ones that are only briefly mentioned and can therefore be written in full.
The following abbreviations have been removed:
CTCs (line 92), VPS (line 128), EVD (line 129), ETV (line 129), TERTp (line 157), TMB (line 295), LOH (line 222), CT (line 43), PCNSL (line 256), SCNSL (line 257).
4- I feel that adding a short overview of how the technique was demonstrated in pre-clinical models of CNS tumours would strengthen the review.
We thank the reviewer for the comment; however, very little amount (few ml in mice) of CSF can be obtained from preclinical models limiting the feasibility of these studies and, hence, there are very few publications studying CSF ctDNA using preclinical models. This is a very interesting concept that can be the subject of another review focusing on the study of liquid biopsies in preclinical models.
5- Line 51: "For intracranial tumours, obtaining a tumour specimen is challenging since surgical resections are invasive and risky procedures." This sentence is redundant since its concept is repeated in the following lines.
This concept is further explained within the same paragraph since it is one of the critical limitations of solid biopsies.
“For intracranial tumours, obtaining a tumour specimen is challenging since surgical re-sections are invasive and risky procedures. To improve disease control, primary brain tumours or single nodule brain metastases are resected. However, obtaining tumour biopsies is not always possible due to their location, particularly when CNS tumours occur in vital regions such as the basal ganglia or the brain stem. In addition, patients with disseminated disease may not be eligible for such procedures [4-6].”
6- Line 63: "This can be challenging with conventional imaging techniques." Please briefly explain the shortcomings of medical imaging in distinguishing true from pseudo tumour progression.
The underlined sentence has been added within the paragraph mentioned by the reviewer.
“It is important to monitor the patient’s response to treatment during the course of the disease, particularly to distinguish a true disease progression from a pseudoprogression induced by treatment. Sometimes a new or enlarging area of contrast enhancement is observed but it is not easy to assess whether it is the result of tumor growth or an inflammatory response [10]. This can be challenging when using conventional imaging techniques.”
7- Please explain the term "the selective pressure of therapy" that appears in line 65.
Therapy can act as a selective pressure by “selecting” the resistant subclones and shaping tumour evolution.
We have added the following explanation (underlined) and a reference of a review that provides more information on the topic:
“Tumours evolve over time, particularly under selective pressure of therapy that can result in the expansion of pre-existing resistant clones or the acquisition of de novo resistant alterations (Dagogo-Jack and Shaw, 2018)”
- Dagogo-Jack and Shaw, 2018. Tumour heterogeneity and resistance to cancer therapies. Nature Reviews Clinical Oncology.
8- Line 99: " The detectable levels of ctDNA in plasma for patients with primary brain tumours may be influenced by the blood-brain-barrier (BBB) integrity". Please briefly elaborate on the selective permeability induced by the BBB and why is tumour DNA impermeable. Importantly, why doesn’t the presence of tumour vasculature, which is much less selective, result in high plasma levels of cell-free, tumour DNA?
We thank the reviewer for raising this point. We have deleted that section from the review.
9- In line 121, please explain the circumstances in which CSF is extracted directly from the ventricles rather than by lumbar puncture? Please add references if possible.
CSF samples are obtained directly from the ventricles in the case of hydrocephalus when an intervention to drain the excess of CSF is required.
This question is clarified within the following paragraphs of the review:
“CSF samples can be accessed through a lumbar puncture or obtained from the ventricles under certain circumstances. Patients with posterior fossa tumours tend to present with hydrocephalus, a condition in which the CSF accumulates within the cerebral ventricles and/or subarachnoid spaces [37-39]. To alleviate the intracranial pressure and to drain the excess of CSF, procedures including ventriculoperitoneal shunt, ventricular drain and endoscopic third ventriculostomy are performed.”
“Obtaining CSF samples is less invasive than surgery; however, in some cases a lumbar puncture is not feasible. Contra-indications to perform a lumbar puncture include: risk for cerebral herniation, abnormal intracranial pressure and coagulation abnormalities [124].”
10- Lines 124-126: " To alleviate the intracranial pressure and to drain the excess CSF, procedures including ventriculoperitoneal shunt (VPS), ventricular drain (EVD) and endoscopic third ventriculostomy (ETV) are performed." This sentence seems irrelevant to the concept of the review.
We consider that this section is necessary since it further clarifies the reviewer’s question 9.
11- Is there a benefit to a liquid biopsy in cases where the tumour can be surgically resected? Or will the solid biopsy provide all the required information? Please elaborate.
Yes, as explained in the text and highlighted in Figures 1 and 2, there are several limitations to solid biopsies, and liquid biopsies can also be used to complement the information obtained from the tumour sample.
12- Please explain the term "longitudinal analysis".
It refers to the analysis performed over follow-up samples collected during a period of time. A graphic description can also be observed in Figure 2.
“During patient follow-up, it could monitor the response to treatment through the levels of ctDNA and the detection of minimal residual disease. Importantly, it could detect early detection of relapse and identify therapeutic targets or mechanisms of resistance to adjust the therapeutic strategy at relapse (Figure 2).”
13- Please explain the term "actionable genomic alterations".
When the gene alteration confers sensitivity or resistance to a clinically available drug.
This has been added in lines 160-163: “In addition, the genomic characterisation of the CSF ctDNA can facilitate the identification of actionable genomic alterations that confer sensitivity or resistance to clinically available drugs, and the detection of mechanisms of resistance at relapse.”
14- In lines 184-186, please clarify that MGMT promoter methylation increases susceptibility to temozolomide. The reader may mistakenly think that MGMT expression does that.
We have further clarified the text (underlined)
“The number of actionable genomic alterations for patients with primary brain tumours is limited. The most relevant biomarker for glioma is the MGMT promoter methylation status. MGMT promoter methylation causes loss of MGMT expression, and since it is involved in DNA repair by reversing DNA alkylation, MGMT promoter methylation renders cells more susceptible to temozolomide and is associated with longer survival [57-60]. MGMT promoter methylation was detected in cfDNA obtained from the CSF of glioma patients with higher sensitivity than the one obtained from serum [61].”
15- Section 4 deals more with ctDNA in the CSF that with the concept of the CSF in general. Therefore, it should be titled otherwise.
We thank the reviewer for raising this point. We have now changed the tittle to: “The cerebrospinal fluid as a source of ctDNA”
16- In lines 255-257, in how many patients was ctDNA found?
The number of patients and variant allele frequencies (VAFs) across the 8 papers is referenced is varied but overall, the CSF is a better source of cfDNA for CNS restricted lymphomas. For example, in Fontanilla et al., plasma ctDNA was found in 8/25 patients with VAFs of 0.5-28.7% and in Hattori et al., serum ctDNA was detected in 8/14 with VAFs of 0.1-0.7% at diagnostic of patients with restricted CNS lymphoma. In contrast, for example in Bobillo et al., CSF ctDNA was detected in 6/6 with VAFs ranging from 9-95% while plasma ctDNA was only detected in 2/6 patients with VAFs < 5%.
17- In line 269, the cited reference is not in the appropriate format.
Thank you, the reference has been introduced in the appropriate format.
Reviewer 4 Report
Escudero and colleagues have written an excellent review on the value of circulating-tumor DNA (ctDNA) in determining the diagnosis and predicting the prognosis and response of brain cancers. The clinical value of ctDNA collected from cerebrospinal fluid (CSF) could be improved by mentioning how it differs from cell-free DNA (cfDNA) which comes from normal cells and is also found in CSF. A description of how ctDNA are detected would highlight its limits in terms of their representativeness of the cancer cell genome.
Minor comment: Define the abbreviations “SOC”, “NGS”
Author Response
We thank the reviewer for revising the manuscript.
The abbreviation “NGS” has been defined in line 105. However, the abbreviation SOC has not been used since we have used the full word (standard of care) three times.
Regarding the clarification of cfDNA vs ctDNA, ctDNA is differentiated from cfDNA by the detection of genomic alterations (mutations, copy number alterations, chromosomal rearrangements).
We have incorporated the following sentence (underlined) and also added a reference that provides more information (Siravegna et al., 2019).
“Cells release DNA that then circulate in bodily fluids. The fraction of cell-free DNA (cfDNA) that is shed by cancer cells, presumably undergoing apoptosis or necrosis, is known as ctDNA and carries genomic alterations that can be detected using PCR-based or next generation sequencing (NGS)-based methods” (REFS +Siravegna et al.,2019) in lines 90-92.
Moreover, as indicated within the text (lines 322-328), techniques used for the analysis of ctDNA include droplet digital PCR (ddPCR), whole exome sequencing (WES), shallow whole genome sequencing (WGS) or specific gene panels.
Round 2
Reviewer 3 Report
Comments 2-4, 6-9 and 12-17 were fully addressed.
However, I require additional consideration to comments 1,5,10 and 11:
Comment 1- maintaining categorization according to type of malignancy is acceptable, although less preferable in my opinion. However, sections 5-8 should still be sub-sections of section 4, since they deal with clinical applications of the technique. It does not make sense to me, to place each application under an independent section as if the concept of the text has changed.
Comment 5- The sentences- (1) "For intracranial tumours, obtaining a tumour specimen is challenging since surgical re-sections are invasive and risky procedures".....and- (2) "However, obtaining tumour biopsies is not always possible due to their location, particularly when CNS tumours occur in vital regions such as the basal ganglia or the brain stem"....are essentially the same. In my opinion, the second one is sufficient to convey the message.
Comment 10- my question in comment 9 was regarding the circumstances in which CSF will be drained directly from the ventricles. The techniques in which such a procedure is performed are irrelevant to the review and add no information to your answer for the previous comment.
Comment 11- the main limitation of solid biopsies given in this review, is the lack of consistent ability to perform them. The authors indicate that liquid biopsies are a suitable substitute. However, the authors do not clearly address the benefits of liquid biopsies in cases where solid biopsies are feasible. A comment on how liquid biopsies can complement solid ones is needed.
Author Response
Comments and Suggestions for Authors (Round 2)
REVIEWER 3
Comments 2-4, 6-9 and 12-17 were fully addressed.
We thank the reviewer for revising the review and our responses to the previous comments
However, I require additional consideration to comments 1,5,10 and 11:
Comment 1- maintaining categorization according to type of malignancy is acceptable, although less preferable in my opinion. However, sections 5-8 should still be sub-sections of section 4, since they deal with clinical applications of the technique. It does not make sense to me, to place each application under an independent section as if the concept of the text has changed.
The sections have been changed according to the reviewer’s comment and the applications for the different CNS cancer types are all now subsections of “Clinical applications of the CSF ctDNA for CNS malignancies”.
Comment 5- The sentences- (1) "For intracranial tumours, obtaining a tumour specimen is challenging since surgical re-sections are invasive and risky procedures".....and- (2) "However, obtaining tumour biopsies is not always possible due to their location, particularly when CNS tumours occur in vital regions such as the basal ganglia or the brain stem"....are essentially the same. In my opinion, the second one is sufficient to convey the message.
Following the reviewer’s suggestion, the first sentence (1) has been removed.
Comment 10- my question in comment 9 was regarding the circumstances in which CSF will be drained directly from the ventricles. The techniques in which such a procedure is performed are irrelevant to the review and add no information to your answer for the previous comment.
CSF samples are obtained from the ventricles in patients that present hydrocephalus and the excess CSF needs to be drained. This is something that could be common in patients with posterior fossa tumours where the CSF can accumulate within the cerebral ventricles and/or subarachnoid spaces.
This has been further clarified in the text as follows (underlined) and references have also been included. Further information about the contraindications of performing a lumbar puncture are explained in lines 335-338.
“CSF samples can be accessed through a lumbar puncture or obtained from the ventricles under certain circumstances. Patients with posterior fossa tumours tend to present with hydrocephalus, a condition in which the CSF accumulates within the cerebral ventricles and/or subarachnoid spaces [37-39]. A lumbar puncture is contraindicated in these patients given the risk of brain herniation; therefore, CSF is obtained from the ventricles during procedures that are performed to alleviate the intracranial pressure and to drain the excess of CSF [38, 40-42]., procedures including ventriculoperitoneal shunt, ventricular drain and endoscopic third ventriculostomy are performed.”
Comment 11- the main limitation of solid biopsies given in this review, is the lack of consistent ability to perform them. The authors indicate that liquid biopsies are a suitable substitute. However, the authors do not clearly address the benefits of liquid biopsies in cases where solid biopsies are feasible. A comment on how liquid biopsies can complement solid ones is needed.
In this review, the limitations of solid biopsies that have been highlighted include the feasibility to perform them, the lack of information since it may not represent the intratumour heterogeneity and that they provide a static snapshot. On the contrary, some of the advantages of liquid biopsies include that they are less-invasive procedures, that CSF is already obtained as standard of care for some patients, that it can represent the tumour heterogeneity, and that it facilitates longitudinal monitoring. Even if a solid biopsy is obtained, the analysis of CSF ctDNA can complement the tumour characterisation (including intratumour heterogeneity) to provide diagnostic and prognostic information and facilitate patient monitoring by assessing response to treatment, early detection of relapse and characterisation of the relapsed tumour (including resistance).
All these applications have been explained throughout the review and summarised in Figures 1 and 2.
Following the reviewer’s advice, the following sentence (underlined) has been added for further clarification within the conclusion’s paragraph (lines 374-375).
“A liquid biopsy of the CSF could characterise the tumour for diagnosis and to provide prognostic information, also complementing the information obtained from the tumour sample if a biopsy or resection was feasible (Figure 1). During patient follow-up, it could monitor the response to treatment through the levels of ctDNA and the detection of minimal residual dis-ease. Importantly, it could facilitate early detection of relapse and identify therapeutic targets or mechanisms of resistance to adjust the therapeutic strategy at relapse (Figure 2).”